# Is the Retinol-Binding Protein 4 a Possible Risk Factor for Cardiovascular Diseases in Obesity?

**DOI:** 10.3390/ijms21155229

**Published:** 2020-07-23

**Authors:** Anna Maria Rychter, Marzena Skrzypczak-Zielińska, Aleksandra Zielińska, Piotr Eder, Eliana B. Souto, Agnieszka Zawada, Alicja Ewa Ratajczak, Agnieszka Dobrowolska, Iwona Krela-Kaźmierczak

**Affiliations:** 1Department of Gastroenterology, Dietetics and Internal Diseases, Poznan University of Medical Sciences, 60-355 Poznan, Poland; piotr.eder@op.pl (P.E.); aga.zawada@gmail.com (A.Z.); alicjaewaratajczak@gmail.com (A.E.R.); agdob@ump.edu.pl (A.D.); krela@op.pl (I.K.-K.); 2Institute of Human Genetics, Polish Academy of Sciences, 60-479 Poznan, Poland; mskrzypczakzielinska@gmail.com (M.S.-Z.); zielinska-aleksandra@wp.pl (A.Z.); 3Department of Pharmaceutical Technology, Faculty of Pharmacy, University of Coimbra, Pólo das Ciências da Saúde, Azinhaga de Santa Comba, 3000-548 Coimbra, Portugal; souto.eliana@gmail.com; 4CEB—Centre of Biological Engineering, University of Minho, Campus de Gualtar, 4710-057 Braga, Portugal

**Keywords:** atherosclerosis, RBP4, cardiovascular disease, obesity, metabolic syndrome, lipoprotein metabolism

## Abstract

Although many preventive and treatment approaches have been proposed, cardiovascular disease (CVD) remains one of the leading causes of deaths worldwide. Current epidemiological data require the specification of new causative factors, as well as the development of improved diagnostic tools to provide better cardiovascular management. Excessive accumulation of adipose tissue among patients suffering from obesity not only constitutes one of the main risk factors of CVD development but also alters adipokines. Increased attention is devoted to bioactive adipokines, which are also produced by the adipose tissue. The retinol-binding protein 4 (RBP4) has been associated with numerous CVDs and is presumably associated with an increased cardiovascular risk. With this in mind, exploring the role of RBP4, particularly among patients with obesity, could be a promising direction and could lead to better CVD prevention and management in this patient group. In our review, we summarized the current knowledge about RBP4 and its association with essential aspects of cardiovascular disease—lipid profile, intima-media thickness, atherosclerotic process, and diet. We also discussed the *RBP4* gene polymorphisms essential from a cardiovascular perspective.

## 1. Introduction

Cardiovascular disease (CVD) constitutes the most common cause of death in European countries, accounting for 2.2 million deaths in females (47% of all-cause of deaths) and 1.9 million deaths in males (39% of all-cause of deaths) [1,2,3,4]. Additionally, it is responsible for 37% and 34% of all years lost (measured by potential years of life lost, PYLL) among females and males, respectively [1]. Not only is CVD a health issue, but it also involves a significant socio-economic impact [1]. It is estimated that by 2030 the total cost of CVD will rise to USD 1044 billion [5]. Currently, more than half (55%) of the costs is derived from direct healthcare (report on Accident and Emergency departments, medications; inpatient-, outpatient-and primary care) and 45% of the costs originate from the informal care and productivity loss due to morbidity and mortality [4]. The major risk factors of CVD development have been identified in the Framingham Heart Study and INTERHEART case-control study [6,7]. Eight risk factors and health behaviors (hypertension, dyslipidemia, diabetes, obesity, smoking, alcohol, diet, sedentary lifestyle) are the World Health Organization’s (WHO) targets for reduction by 2025 [8]. Obesity is a serious health problem and, if the current trends continue, in the next ten years, almost 40% and 20% of the global adult population will be suffering from overweight and obesity, respectively [9]. Currently, it is estimated that over 3 million patients worldwide die due to excessive body weight [10]. Moreover, obesity may influence cardiovascular risk by means of the presence of obesity-related comorbidities, hemodynamic repercussions, body fat mass content and distribution [11,12,13]. All-cause mortality increased log-linearly throughout the overweight range, with the hazard ratio (HR) of 1.39 per 5 kg/m^2^ [14]. Excessive accumulation of adipose tissue, particularly visceral fat (VF), contributes to a higher prevalence of hypertension, dyslipidemia, and glucose intolerance, which lead to CVD development [15,16,17]. Adipose tissue also regulates many systemic and pathological processes due to the secretion of bioactive proteins—adipokines [18]. Most of them—e.g., tumor necrosis factor-α (TNF-α); IL-6: interleukin (IL-6)—have pro-inflammatory properties and have been associated with vascular and atherothrombotic complications in atherosclerosis, since they influence the function of endothelial cells, arterial smooth muscle cells, and macrophages in the vessel walls [19,20,21,22,23]. However, few of them may have a protective effect in CVD (e.g., adiponectin). In fact, among patients with obesity, the secretion of adipokines is frequently abnormal [24]. In this non-systematic review, we have characterized one of the novel adipokines—retinol-binding protein 4 (RBP4), with a particular emphasis on its role in obesity and CVD development. However, there is an ongoing controversy with regard to any RBP4 role in both inflammation and CVD prediction. Thus, more reviews and research studies are still necessary.

## 2. Role and Structure of RBP4

Retinol-binding protein 4 (RBP4), presented in Figure 1, belongs to the lipocalin family and has a tertiary structure known as the ‘lipocalin fold’ which facilitates the binding of small hydrophobic molecules, such as lipids [19]. RBP4 is synthesized and secreted in the liver (mostly) and other tissues, such as adipose tissue [24]. It transports vitamin A (retinol) from the liver to target tissues and constitutes a major regulator of circulating levels of retinol [25]. RBP4 is transferred within the bloodstream in combination with transthyretin (TTR), which prevents kidney filtration and catabolism of RBP4 [26]. The receptor proteins for RBP4 are STRA6 (stimulated by retinoic acid gene 6) and RBPR-2 (RBP4-receptor 2) [26]. In fact, urinary excretion of RBP4 may be a useful marker for the detection of renal dysfunction [27,28]. In patients with chronic kidney disease (CKD), higher serum RBP4 levels were associated with a higher rate of cardiovascular events and higher mortality, which suggests that RBP4 levels may indicate an increased risk of cardiovascular risk within this group [29]. In the Bobbert et al. study, serum RBP4 levels were also higher in diabetic patients compared to nondiabetic individuals, although not in terms of the levels of retinol and transthyretin [30]. Nevertheless, it should be emphasized that RBP4, retinol, and retinoic acid can differentially affect CVD and metabolic diseases [31]. Therefore, vitamin A metabolism should be taken into consideration when investigating the role of RBP4 since it can act by itself or affect retinol metabolism and retinoic acid signaling [32]. Additionally, vitamin A deficiency reduces serum RBP4 levels, and, hence, it is essential to evaluate not only the metabolism of vitamin A but also its dietary intake when assessing RBP4 concentrations [33]. A decrease in adipose tissue GLUT4 expression, which is the major glucose transporter protein mediating glucose uptake, leads to increased serum RBP4 levels associated with the induction of insulin resistance in the liver and muscle [34]. It has been shown that moderate weight reduction lowers serum RBP4 levels in nondiabetic subjects [27]. However, new evidence suggests that RBP4 plays a more significant role in the lipid metabolism than in insulin resistance [35]. One of the questions regarding RBP4 is which of its serum concentrations are normal, and which are pathological. In healthy individuals, normal serum RBP4 ranges from 10 to 50 µg/mL (without vitamin A deficiency), but among individuals with Type 2 diabetes, CVD or obesity, it may reach up to 150 µg/mL [26,30,33,34,36,37,38,39]. In the study led by Farjo et al., the most substantial influence on pro-inflammatory molecules was achieved with RBP4 serum concentrations of 100 µg/mL [40]. However, lower concentrations in the range of 10 to 25 µg/mL were also shown to be enough to influence pro-inflammatory molecules, which suggest that endothelial cells may be responsive to even small elevations in serum RBP4 concentrations [33,40,41]. RBP4 is also known as a negative acute-phase reactant, and hospitalization may decrease its serum levels [33]. Therefore, the choice of the assay employed in the measurement of serum RBP4 levels should also be careful. As Graham et al. presented in their study, quantitative Western blotting is the most reliable method for assaying serum RBP4 elevations associated with insulin resistance [39]. However, other measurement methods—e.g., ELISA (enzyme-linked immunosorbent assay), EIA (enzyme immunoassay)—are also widely used in various populations [42]. Additionally, RBP4 concentrations were found to be different among men and women. Some research studies indicated that RBP4 was significantly higher in men than in women, including adolescent boys and girls, whereas no such association was observed in other studies [35,43,44]. This could be explained by different amounts and distribution of the adipose tissue (including liver fat), the influence of the sex hormones, and iron metabolism [44,45,46]. RBP4 levels were also markedly different between premenopausal and postmenopausal healthy women, with higher levels among the second group [47]. The summary of measurement methods and serum RBP4 range in the selected studies, according to CVD risk assessment, are listed in Table 1. 

## 3. *RBP4* Gene—Structure and Polymorphism vs. CVD in Obesity

RBP4 protein is encoded by the same name gene—*RBP4* (MIM 180250) located on chromosome 10 (10q23.33) between coordinates 93,591,694 and 93,601,744 bp according to human genome reference assembly GRCh38.p13 (Figure 2). It encompasses 10,050 bp of genomic DNA and consists of 6 exons, including five coding fragments. Transcript length is 1070 bp, and the translation product consists of 201 amino acids.

Since RBP4 may be correlated with conditions related to Type 2 diabetes mellitus (T2DM), obesity, or CVD, the *RBP4* gene may constitute the gene carrying obesity-related implications [54]. Additionally, the *RBP4* gene location is close to a region linked with an increased risk of T2DM and elevated fasting blood glucose levels [55,56,57]. Thus, it has been proposed that functional *RBP4* gene polymorphisms influence a higher obesity incidence, insulin resistance, hyperinsulinemia, T2DM, and artery thickness [26]. These hypotheses were confirmed by research carried out in recent years by several research teams. The most significant *RBP4* gene variants connected with CVD and its markers are summarized in Table 2 and presented in Figure 2, with the distribution of each *locus* and the minor allele frequency (MAF) occurrence in the world population based on 1000 Genomes Project data (phase 3; https://www.internationalgenome.org/). It is worth noting that all of the listed variants are located outside the coding gene region and that there can be a potential relationship between the regulation and the change in gene expression levels.

## 4. RBP4, Obesity, and Metabolic Syndrome 

As mentioned before, the secretion of adipokines is frequently abnormal among patients with obesity. Adipocyte hypertrophy, ectopic fat accumulation, and adipose tissue inflammation may cause adverse adipokine secretion, which, in turn, can be associated with a number of health consequences, including metabolic, inflammatory, or cardiovascular diseases [63]. However, in several studies, RBP4 levels were higher among individuals with obesity in comparison to control groups. In other studies, no such correlation has been found. In the Korek et al. study, RBP4 levels did not correlate with BMI or fat mass and did not differ between individuals with obesity and those without obesity—RBP4 levels in both groups were 33.93 ± 4.46 and 32.53 ± 2.53 µg/mL, respectively [64]. Similar results were reported by other authors [65,66,67,68]. On the other hand, certain studies demonstrate increased RBP4 concentrations among individuals with obesity, as well as the association between RBP4 and BMI [36,69]. Therefore, it has been suggested that RPB4 concentrations may not be related necessarily to obesity itself, but to the location of the adipose tissue. The expression seems to be higher in visceral (VF) than in the subcutaneous tissue (SF); thus, RBP4 levels are more closely associated with VF levels and appear to constitute the best indicator of intra-abdominal adipose mass [35,70,71]. RBP4 may be the mechanistic link between the visceral adiposity and increased cardiovascular risk associated with this type of adipose tissue [49,71]. In the study by Lee et al., RBP4 levels were correlated with visceral fat areas, but not with the total body fat (wt.%), and, as the authors suggested, RBP4 could be the link between visceral obesity and atherosclerotic vascular changes [72]. Furthermore, RBP4 levels may also be prone to weight loss. In fact, serum RBP4 levels decreased considerably by 25.5% after weight reduction—almost 11% of weight loss in the course of a 16-week program [27]. However, it is essential to point out that in addition to a reduced caloric intake by 600 kcal/day, sibutramine was also used. Interestingly, the statistically significant increase in RBP4 levels was also observed in patients undergoing bariatric surgery—RBP4 levels (ng/mL) were 22,456.5 ± 13,158.8 and 31,342.2 ± 8172.5 in pre-and post-bariatric periods, respectively [68]. In two different studies, RBP4 levels decreased significantly following bariatric surgery [69,70]. According to Zachariah et al., participants with serum (log-transformed) RBP4 levels at the 4th quartile presented a 75% higher risk of developing the metabolic syndrome when compared to patients in the 1st quartile [73]. Moreover, in the study by Karamfilova et al., RBP4 levels ≥55 mcg/mL were associated with a 3.1 higher risk of developing metabolic syndrome [74]. Other studies also have confirmed the relationship between the components of metabolic syndrome and RBP4 levels [65,66]. Additionally, RBP4 can also be a predictor for the diagnosis of metabolic syndrome and weight regain [70,74,75]. In fact, Vink et al. demonstrated that RBP4 was a predictor of weight regain—stronger in men and individuals following a low-calorie diet than in women and individuals following a very-low-calorie diet [75]. 

Possible explanations regarding different concentrations or changes in RBP4 may include both ethnic and age differences (e.g., presence of renal dysfunction) [66]. Despite different data, there is still a strong association between RBP4 and obesity.

## 5. RBP4 and Lipid Metabolism 

RBP4 and retinoids are involved in the lipid status since they influence the metabolism of triglycerides [26]. RPB4 levels are associated with dyslipidemia, which is a known risk factor for atherosclerosis. Strong, positive correlations between RBP4 levels and triglycerides, which constitute two major lipid abnormalities in both T2D and metabolic syndrome individuals, were observed in patients with and without obesity [64]. Higher RBP4 levels were correlated with higher levels of triglycerides and lower levels of high-density lipoprotein (HDL) cholesterol [66,76], as shown in Figure 3. Rocha et al. demonstrated that triglycerides were an independent predictor for RBP4 levels [35]. High RPB4 levels may be involved in the pro-atherogenic plasma lipoprotein profile. In T2DM patients, RBP4 and retinol were positively correlated with triglycerides, total cholesterol, apoB, and non-HDL and low-density lipoprotein (LDL)-cholesterol [65]. Additionally, positive univariate correlations were observed with LDL-P, very-low-density lipoprotein (VLDL)-P, small LDL and HDL, and large and medium VLDL (including chylomicrons if present) [65]. According to Ingelsson et al., RBP4 levels were poorly correlated with the total cholesterol and triglycerides and were not associated with HDL-and LDL-cholesterol [37]. Conversely, in other studies, RBP4 levels were not statistically significant with respect to lipid metabolism [19,52].

Although RBP4 may be involved in different periods of the atherosclerotic process, its role in lipid metabolism remains unclear and should constitute the focus of further research. 

## 6. RBP4 and the Endothelium

Chronic vascular inflammation plays a critical role in the development of atherosclerosis [77]. It begins with the endothelial secretion of pro-inflammatory cell surface adhesion molecules and soluble pro-inflammatory factors—endothelial-leukocyte adhesion molecule (E-selectin), intercellular adhesion molecule 1 (ICAM-1), vascular cell adhesion molecule 1 (VCAM-1), interleukin-6 (IL-6), monocyte chemoattractant protein 1 (MCP-1) [18,42]. RBP4 may also be involved in oxidative stress and in the initiation of endothelial inflammation [33,42,78]. In T2DM patients, RBP4 levels were positively correlated with sICAM-1 and sE-selectin, which were related to the progression of vascular complications associated with diabetes [76]. In the same study, RBP4 levels were significantly and strongly negatively correlated with the flow-mediated vasodilatation (FMD). However, this study included a population with newly diagnosed T2DM without the use of any medications which could potentially affect the endothelial function. The negative correlation between RBP4 and FMD may reflect the endothelial function as a result of nitric oxide (NO production, which is a major vasodilatory substance in the endothelium) [79]. As Takebayashi et al. study showed, RBP4 has a major effect of increasing NO production due to the stimulation of the part of the PI3K/Akt/eNOS pathway and the inhibition of ERK1/2 phosphorylation and insulin-induced ET-1 secretion leading to vasodilatation [79]. Elevated RBP4 levels may contribute to sustaining or initiating the pro-inflammatory status by activating macrophages, and it is mediated partially through the c-Jun-N-terminal protein kinase (JNK) and Toll-like receptor 4 (TRL4) pathways, independent of retinol-binding to RBP4 and STRA6 [80,81]. However, the RBP4 effects are not fully blocked in TLR4^-^/macrophages, which suggests that alternative pathways may be considered [80]. In the study by Farjo et al., RBP4-mediated endothelial inflammation was also independent of retinol and STRA6 and was acting via NADPH oxidase-and NF-β-dependent pathways [40]. Moreover, RBP4 induced expression and secretion of pro-inflammatory cytokines in macrophages, including MCP-1, TNF-α, IFN-γ (interferon-gamma), IL-6, IL-2, IL-1β, IL-12p70, GM-CSF (granulocyte macrophage-colony stimulating factor) [80]. On the other hand, RBP4 signaling in adipocytes was retinol and STRA6 dependent [82]. Another study showed that serum RBP levels were independently and inversely associated with E-selectin in rheumatoid arthritis (RA) patients aged ≤55 years with two or more traditional CV risk factors, abdominal obesity and RA of over ten years’ duration [83]. RPB4 levels were not associated with VCAM-1, ICAM-1, and MCP-1 levels. 

As the research studies have demonstrated, RBP4 is associated with insulin resistance (IR), which is further associated with chronic subclinical inflammation and can promote vascular inflammation [36,84,85,86,87]. Moreover, RBP4 can induce IR by developing the inflammatory state in the adipose tissue due to the activation of pro-inflammatory cytokines in macrophages. Holo-RBP4 (not bounded to the retinol) induces IR by binding to the STRA6 in the adipocytes through JNK. RBP4 also suppresses insulin signaling by inducing suppressor of SOCS3 (suppressor of cytokine signaling 3) [26,42]. Additionally, it can inhibit insulin signaling by means of the activation of TLR4 pathways independent of the STRA6 [80]. Either RBP4 bounded (apo-RBP4), or not bounded to the retinol could induce IR and, consequently, endothelial inflammation. In the Framingham Heart Study (third generation cohort), increased levels of RBP4 and fetuin-A were observed, which was associated with the incidence of metabolic syndrome, regardless of obesity [73]. These results could suggest that the association between adverse adipokine profile and the incidence of metabolic syndrome is the result of overlaying other mechanisms, such as insulin resistance. Similarly, circulating RBP4 levels predicted the development of metabolic syndrome and its components, including IR, in adolescents, irrespective of obesity [88]. As the authors suggested, this mechanism is strongly associated with RBP4 as an indicator of IR. According to Jialal et al., the RBP4/adiponectin ratio correlated significantly with a high-sensitivity CRP (C-reactive protein), but not with HOMA-IR [89]. However, other studies have failed to find the association between insulin resistance and RBP4; therefore, the question of whether RBP4 is the causative factor, or the result of the insulin resistance should be investigated further. 

## 7. RBP4 and Intima-Media Thickness

As has already been mentioned, the prevalence of CVD, of which atherosclerosis is the major component, is still increasing. There is a strong need for undertaking both standard and novel diagnostic methods among men and women for an early prevention plan to be implemented. Identification of atherosclerosis at the subclinical stages would primarily promote earlier diet prevention and a selection of more effective treatment methods, which may lead to a better prognosis [19]. Ultrasonography is more common, non-invasive, and one of the most effective methods in the diagnosis of early structural changes in the artery wall, even before the disease symptoms are present [90]. The intima-media thickness is the distance from the lumen–intima interface to the media–adventitia interface of the artery wall, as measured in noninvasively obtained ultrasonographic images of the carotid arteries. Carotid-wall intima-media thickness (cIMT/IMT) is one of the methods used to measure atherosclerosis associated with both cardiovascular factors and their outcomes [91,92]. In addition, IMT measurement is of low-cost, and is performed without the need to use a contrast medium, and is characterized by high recurrence as well as high-quality imaging [93].

Although the intima-media thickness measurement is a well-established method, its correlation with serum RBP4 levels is not quite well understood and remains undefined. To evaluate whether there is any interaction between these two parameters, further research is still necessary. In the Mansouri et al. study, there was no correlation between cIMT and RBP4 levels in Type 2 diabetes patients (T2DM) [50]. Similar result was reported by other authors [33,51,67,76,94]. In the study conducted by Huang et al., it was observed that both low and high RBP4 levels could be associated with coronary artery calcification [67]. On the other hand, according to Feng et al., a high RBP4 level was one of the seven factors associated with the elevated CIMT, and, as the author suggested, it could be used as an early predictor of CVD in Type 2 diabetes patients [48]. Comparable results were reported with regard to serum RBP4 and lipocalin-2 levels in newly diagnosed T2DM [19]. In both type 1 and type 2 diabetes patients, the use of glucose-lowering medications, or their combinations, could affect IMT either positively or negatively [95,96,97,98,99]. Burchardt et al. claimed that the adjunctive use of metformin in type 1 diabetes patients led to a reduction of the maximum cIMT after six months, in contrast to patients receiving only insulin whose IMT increased [100]. However, similar results were not found in T2DM patients during the Copenhagen Insulin and Metformin Therapy trial [101]. Additionally, some studies questioned whether glucose-lowering medications would also affect RBP4 levels. In the polycystic ovary syndrome patients, simvastatin alone, or with metformin, did not affect serum RBP 4 levels [102]. On the other hand, in an animal study conducted by Abbas et al., metformin, liraglutide, and sitagliptin decreased serum RBP4 levels following eight weeks of treatment [103]. Moreover, RBP4 levels were also associated with IMT among women with untreated essential hypertension [38]. In the study by Bobbert et al., retinol was inversely correlated with IMT [30]. In addition, the retinol/RBP4 ratio indicating the saturation of RBP4 with retinol was strongly associated with intima-media thickness. Thus, it suggests that retinol-free RBP4 may be involved in the atherosclerosis process. The potential pathomechanism of RBP4’s impact on atherosclerosis and CVD risk is presented in Figure 4. 

## 8. Diet and Its Influence on RBP4 Levels

Nutritional and lifestyle care are some of the essential approaches to CVD management and prevention. Standard dietary patterns, such as the Dietary Approach to Stop Hypertension (DASH) or Mediterranean diet (MeD), as well as the new plant-based diets, are recommended in CVD risk reduction [11]. The question of whether nutritional compounds, particularly anti-inflammatory, or the general lifestyle features can affect RBP4 levels should be addressed when considering the possible association between RBP4 and CVD. 

Compliance with the DASH diet was independently associated with lower RBP4 serum levels among middle-aged and elderly adults [104]. Furthermore, higher compliance with the MeD resulted in lower RBP4 levels, regardless of weight loss or caloric restriction [105]. On the other hand, a high-quality, plant-based diet was associated with lower plasma levels of several adipokines, but not with RBP4 levels and inflammatory markers [106]. Similar results were shown in other studies [107,108]. It has been suggested that protein intake is more crucial than the caloric intake for RBP4 levels [109]. In fact, an energy-restricted diet with a higher protein intake was associated with a 30% greater decrease in RBP4 levels (35% compared with 20% of protein intake) [110]. In another study, serum RBP4 levels decreased more in the course of a hypocaloric carbohydrate-restricted diet than during a hypocaloric low-fat diet [111]. However, in both diets, changes in RBP4 were associated with changes in LDL particle size—essential in the atherosclerotic process and associated with IMT. According to Daneshzad et al., RBP4 levels were positively associated with vitamin A intake among patients with obesity [112]. This result could be explained by the fact that a higher vitamin A intake would require higher RBP4 levels for transportation, storage, and metabolization. As the Zhou et al. study showed, RBP4 was a predictor for developing diabetic atherosclerosis in an animal model, and its levels decreased following a vitamin D supplementation [113]. Omega-3 supplementation decreased RBP4 levels among adolescents with obesity; however, the effect was not statistically significant when compared to lifestyle intervention alone [114]. It has been shown that the intake of selenium, an antioxidant mineral, is inversely associated with RBP4 levels [115]. Nevertheless, data concerning weight-loss and its influence on the changes in RBP4 levels are contradictory, and this aspect should be investigated further [70,105,109]. However, it is essential to keep in mind that weight loss does not have to affect the hepatic production of RBP4, although it may impact adipose tissue RBP4-production. Interestingly, RBP4 possibly correlates with an increased regain of lost weight and is one of the predictors of metabolic syndrome among people with excessive body weight [70,75]. Lifestyle features, such as smoking or physical activity, were associated positively and negatively with plasma RBP4 levels, respectively [115,116]. The lifestyle factors affecting RBP4 levels are schematically presented in Figure 5.

## 9. Can RBP4 Be Used as a Biomarker in CVD?

Current studies are focused on finding new biomarkers which would provide an added value to the well-known CVD risk factors and could serve as a diagnostic tool. Retinol-binding protein 4 should be investigated from the biomarker perspective since RBP4 levels were observed to be higher among individuals with CVD and could be involved in the atherosclerotic process. 

According to Alkharfy et al., serum RBP4 levels correlated significantly well with the existing risk factors of cardiovascular disease in women, regardless of body weight [117]. This result could suggest that RBP4 measurement in women can be a reliable predictor of the developing CV events, and could provide prognostic knowledge. The comparable result was obtained in the study by Pala et al., where RBP4 levels were statistically increased among individuals with incidental fatal or nonfatal ischemic heart disease, or the cerebrovascular disease, when compared to the control group [118]. Furthermore, the RBP4/adiponectin ratio was significantly increased among individuals with nascent metabolic syndrome and, thus, could constitute a predictor of CVD in this patient group in large, prospective studies [89]. RBP4 was also up-regulated in the pre-term group of neonates, and, as the authors suggested, it should be included in the detection of neonates at higher risk of developing CVD [119]. Cabré et al. suggested that type 2 diabetic participants with RBP4 plasma levels in the fourth quartile presented an over two and a half-fold increased risk of developing CVD, irrespective of bad metabolic control [28]. On the other hand, Kim et al. concluded that RBP4 could be used as a diagnostic marker of CVD among non-diabetic individuals [120]. Opposite results were obtained by Patterson et al., who reported that higher RBP4 levels were associated with a decreased risk of non-CVD mortality [121], whereas Liu et al. showed that RBP4 levels were inversely (but not significantly) associated with CVD mortality [122]. 

## 10. Conclusions

Retinol-binding protein 4 is one of the adipokines potentially associated with an increased risk of developing cardiovascular disease, particularly among patients with obesity. The role of RBP4 in the atherosclerotic process is mainly associated with an increased expression of pro-inflammatory cell surface adhesion molecules and soluble pro-inflammatory factors. RBP4 has also been connected with an unfavorable lipid profile and an increased intima-media thickness; however, these associations should constitute the focus of further extensive research. To investigate the role of RBP4 in CVD more precisely, other factors, such as diet or *RBP4* gene polymorphisms, should also be included in the extensive studies.

It is difficult to determine at this point whether RBP4 could constitute a novel biomarker useful in CVD and which pathological process it could represent (e.g., inflammation, metabolic or oxidative stress). Additionally, further research should be conducted to confirm whether RBP4 is yet another risk factor of CVD, as RBP4 can represent a promising indicator in the treatment and diagnosis of CVD.

## Figures and Tables

**Figure 1 ijms-21-05229-f001:**
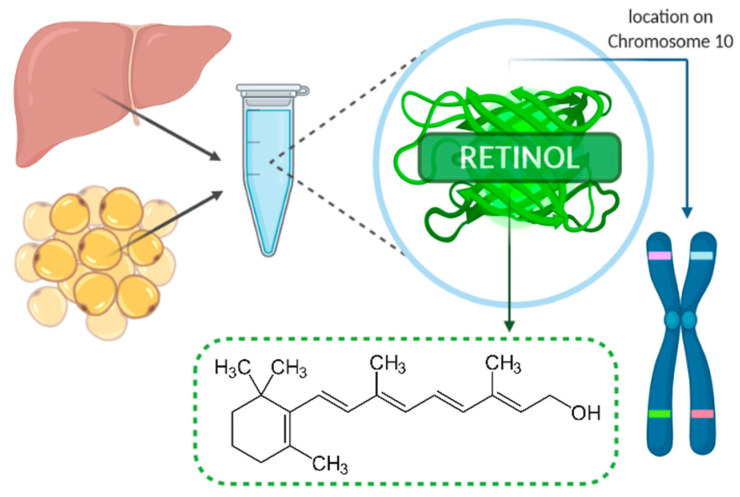
Retinol-binding protein 4 (RBP4).

**Figure 2 ijms-21-05229-f002:**
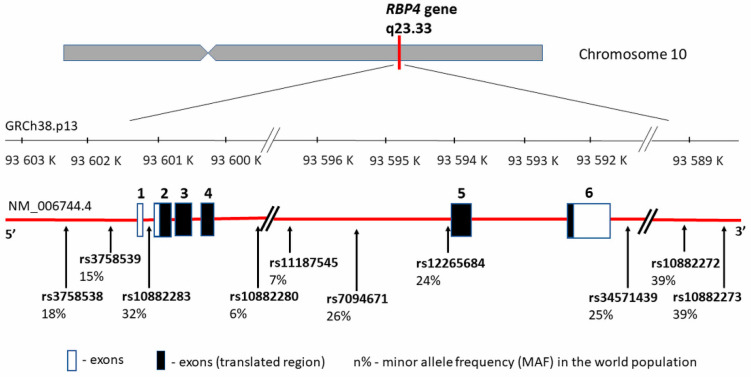
*RBP4* gene structure, chromosome location, and cardiovascular disease (CVD) variants distribution. rs—number of the reference sequence in the National Center of Biotechnological Information database.

**Figure 3 ijms-21-05229-f003:**
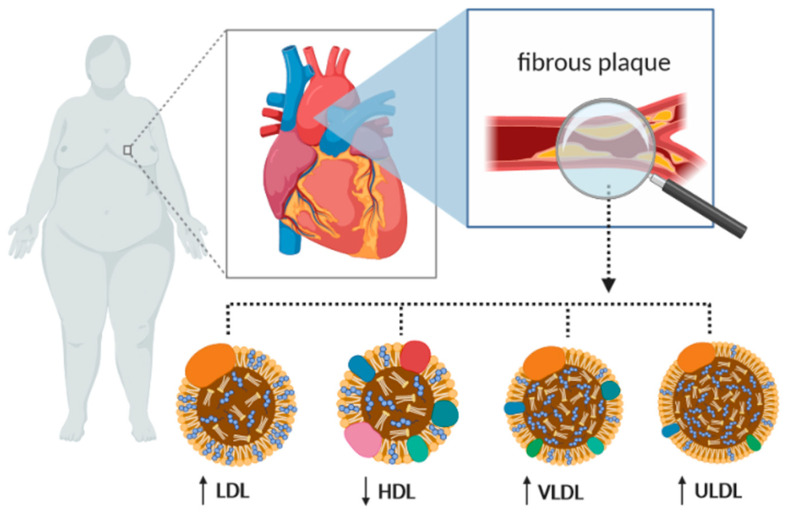
Lipid disorders impacted by RBP4 as the main levels of risk of cardiovascular disease caused by obesity. LDL—low-density lipoprotein; HDL—high-density lipoprotein; VLDL—very-low-density lipoprotein; ULDL—ultra low-density lipoprotein.

**Figure 4 ijms-21-05229-f004:**
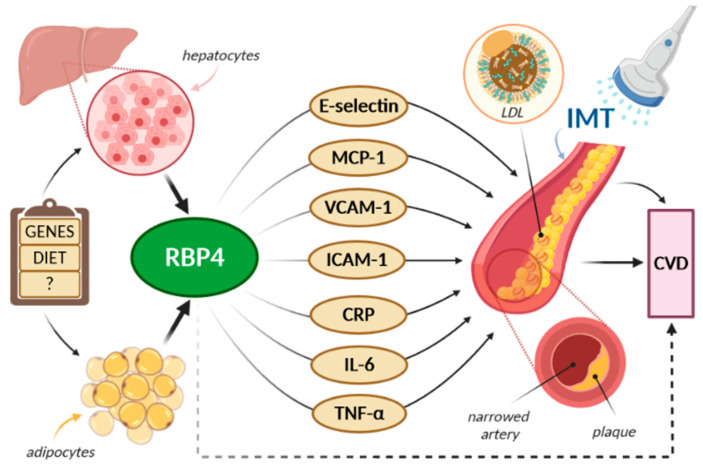
The potential pathomechanism of RBP4’s impact on atherosclerosis and CVD risk. E-selectin: endothelial-leukocyte adhesion molecule; MCP-1: monocyte chemoattractant protein 1; VCAM-1: vascular cell adhesion molecule 1; ICAM-1: intercellular adhesion molecule 1; CRP: C reactive protein; IL-6: interleukin-6; TNF-α—tumor necrosis factor α.

**Figure 5 ijms-21-05229-f005:**
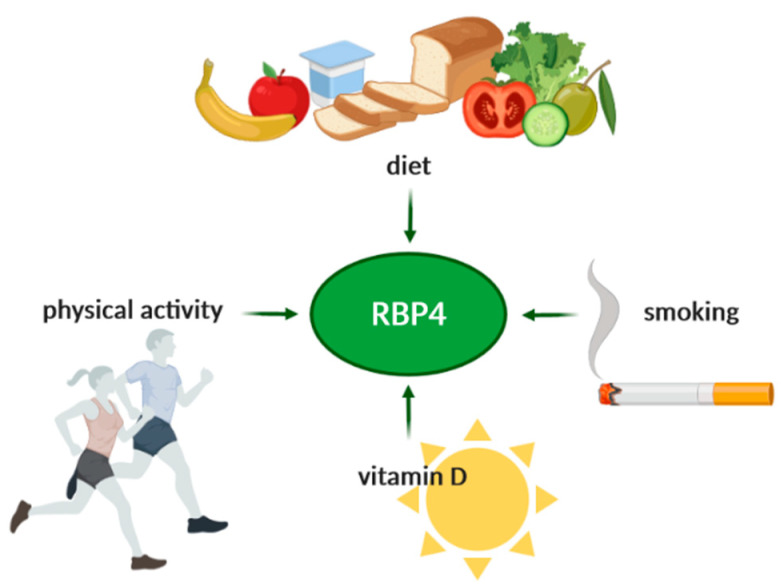
Diet, lifestyle, and RBP4.

**Table 1 ijms-21-05229-t001:** The summary of measurement methods and serum retinol-binding protein 4 (RBP4) range in studies assessing cardiovascular (CV) risk.

Authors	Study Population	Groups, Sex (Group Size, *n*)	Age (Years)	BMI (kg/m^2^)	CV Risk Assessment Method	CV Risk (Group Size, *n*)	RBP4 Measurement Method (Unit)	RBP4 Specimen	Serum RBP4 Range	Relation between RBP4 and CV Risk
Feng et al. 2015 [48]	T2DM	498 F; 578 M (1076)	62.80 ± 13.60	27.50 ± 4.20	cIMT (mm)	G 1 (332): no abnormalities	ELISA (mg/L)	serum	G 1 32.10 ± 10.3	+
				27.90 ± 3.40		G 2 (386): ≥ 1			G 2 38.20 ± 8.30	
				27.60 ± 3.60		G 3 (358): ≥ 1.5			G 3 46.90 ± 7.60	
Xiao et al. 2013 [19]	T2DM	140 F; 144 M (284)	35.00–70.00	25.10 ± 2.80	cIMT (mm) fIMT (mm) iIMT (mm)	subAS (78) cIMT 0.94 ± 0.34 fIMT 0.97 ± 0.33 iIMT 1.13 ± 0.28	ELISA with monoclonal antibodies (mg/L)	serum	37.1 (32.3–40.8)	+
				24.50 ± 2.80		Non-subAS cIMT 0.70 ± 0.11 fIMT 0.70 ± 0.11 iIMT 0.76 ± 0.10			23.2 (20.1–29.2)	+
Won et al. 2012 [49]	Healthy	175 F; 116 M (291)	40.00 ± 11.00	27.00 ± 2.60	The Framingham Risk Score	MetS (57) Framingham risk: 2.0, 0.0 to >30.0 Framingham score: 9.0, −7.0 to 17.0	EIA (µg/mL)	plasma	MetS 65.1 ± 26.8	+
				23.60 ± 3.00		Non-MetS (234) Framingham risk: 0.5, 0.0 to 20.0 Framingham score: 3.0, −9.0 to 18.0			Non-MetS 52.2 ± 20.0	
Su et al. 2020 [29]	CKD	58 F; 111 M (169)	59.50–78.00	27.40 ± 2.90	CV events (fatal and nonfatal)	(total 80) CV events: 41 CV mortality: 10	ELISA (mg/L)	serum	>33.86	+ (higher rates of CV events than RBP4 < 33.86)
				25.90 ± 2.10		(total 89) CV events: 11 CV mortality: 4			<33.86	+
Solini et al. 2009 [38]	HYP	35 F	47.40 ± 5.00	25.00 ± 1.60	cIMT (mm)	0.54 ± 0.15	ELISA (µg/mL)	plasma	Median value 38.75	+
	CTL	35 F	46.90 ± 6.30	25.70 ± 1.40		0.5 ± 0.13			Median value 10.00	None
Mansouri et al. 2012 [50]	T2DM	53 F; 48 M (101)	53.60 ± 8.40	27.70 ± 4.10	cIMT (mm)	0.8 ± 0.2	ELISA (µg/mL)	serum	71.9 ± 35.6	None
Bobbert et al. 2010 [30]	T2DM and non-T2DM	52 F; 44 M (96)	55.00 ± 1.30	30.80 ± 0.70	cIMT (mm)	0.72 ± 0.02	ELISA (µmol/L)	serum	1.89 ± 0.05	+
Chu et al. 2011 [51]	T2DM with CKD	86 (sex NM)	70.00 ± 11.00	26.20 ± 6.20	cIMT (mm)	0.75 ± 0.16	ELISA (µg/mL)	serum	44.8 ± 6.4	None
	T2DM without CKD	153 (sex NM)	60.00 ± 12.00	26.30 ± 5.90		0.69 ± 0.14			39.5 ± 4.9	None
Li et al. 2020 [52]	CHF	227 F; 707 M (934)	≥60	22.49–26.67	MACE (Multivariable Cox regression)	-	ELISA (µg/mL)	serum	46.66 ± 12.38	+ (log RBP4 associated with 1.6 times higher risk of MACE)
Bachmayer et al. 2013 [53]	Patients with obesity	65 F; 27 M (92)	43.00 ± 10.00	50.00 ± 7.00	Endothelial dysfunction: CRAE (µm); CRVE (µm); AVR	CRAE 178 ± 19	ELISA (ng/mL)	NM	24,773 ± 14,025	None
						CRVE 221 ± 24				None
						AVR 0.81 ± 0.09				None

F—female, M—men, ± SDs, T2DM—Type 2 diabetes mellitus, G—group, ELISA—enzyme-linked immuno-absorbent assay, CV—cardiovascular, IMT—intima-media thickness, cIMT—carotid intima-media thickness, fIMT—femoral intima-media thickness, iIMT—common iliac intima-media thickness, subAS—subclinical atherosclerosis, EIA—enzyme immunoassay, MetS—metabolic syndrome, CKD—chronic kidney disease, HYP—hypertensive, CTL—normotensive, +—positive, NM—not mentioned, CHF—chronic heart failure, MACE—major adverse cardiac event(s) (cardiovascular death and rehospitalization due to the deterioration of CHF), CRAE, CRVE—central retinal artery/vein equivalent, AVR—arterio–venous-ratio.

**Table 2 ijms-21-05229-t002:** *RBP4* gene variants investigated as risk factors of cardiovascular diseases in obesity.

Variant	Genetic Location	Study Group		Pathophysiology Association	Reference
		*n* (Total)	Diagnosis		
rs10882280	g.6681G > T c.355+837G > T (intronic)	1422 F; 414 M (1836)	healthy (metabolic, cardiovascular, or endocrine disease excluded)	Higher high-density lipoprotein level associated with minor allele T (*p* = 0.043) and C (*p* = 0.042), respectively	Shea et al. 2010 [58]
rs11187545	g.8889T > C c.355+3045T > C (intronic)				
rs10882283	g.5030T > G c. −55T > G (5’ UTR variant)	457 F; 477 M (934); 716 CTL	T2DM	G-allele associated with a higher body-mass index and waist-to-hip ratio values (*p* < 0.05).	Kovacs et al. 2007 [59]
rs10882273	g.27484T > C c.*1539T > C (3′ UTR variant)	457 F; 477 M (934); 716 CTL	T2DM	C-allele associated with an increased BMI, plasma insulin, and circulating free fatty acid concentrations (*p* < 0.05)	Kovacs et al. 2007 [59]
		1787 F;1423 M (3210)	Chinese Hans population 50–70 years old	Higher body-mass index values. Higher insulin and free fatty acids levels. Association with plasma RBP4 levels (*p* = 0.005).	Wu et al. 2009 [60]
rs10882272	g.26761T > C c.*816T > C (3′ UTR variant)	593 F; 454 M (947)	French-Canadian founder population 12–18 years old	Association with circulating retinol levels. Modulation between vitamin A intake and abdominal adiposity.	Goodwin et al. 2015 [61]
		5 006	Caucasian cohorts from Finland, USA, and Italy	Association with circulating retinol levels.	Mondul et al. 2011 [25]
rs3758538	g.3944A > C c.697–1781A > C (upstream transcript variant)	97 with obesity; 83 normal-weight	Spanish Caucasian children	Association with triglycerides levels and plasma RBP4 levels. C allele associated with obesity and higher BMI z-score.	Codõner-Franch et al. 2016 [54]
		1787 F; 1423 M (3210)	Chinese Hans population 50–70 years old	Association with hypertriglyceridemia and plasma RBP4 levels.	Wu et al. 2009 [60]
rs3758539	g.4406G > A c.697-2243G > A (upstream transcript variant)	97 cases 83 CTL	Obesity, Spanish Caucasian children	Association with triglycerides levels in children.	Codõner-Franch et al. 2016 [54]
		66 F; 63 M (129) 192 CTL	Obesity, cohort from Iran	Association with an increased susceptibility for obesity and an increased BMI.	Shajarian et al. 2015 [55]
rs12265684	g.12177G > A c.356-25G > A (intronic)	97 cases 83 CTL	Obesity, Spanish Caucasian children	Association with triglycerides levels and blood pressure.	Codõner-Franch et al. 2016 [54]
rs34571439	g.14684T > G c.697-12521A > C (upstream transcript variant)			Association with triglycerides and plasma RBP4 levels as well as plasma C-reactive protein values.	
rs7094671	g.10377C > T c.356-1825C > T (intronic)	297 M; 217 M CTL	CAD, Chinese patients	G allele associated with a higher risk of CAD	Wan et al. 2014 [62]

rs—number of the reference sequence in the National Center of Biotechnological Information database, UTR—untranslated region, F—female, M—men, CTL—controls, T2DM—Type 2 diabetes mellitus, CAD—coronary artery disease.

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
