# Peer review of "Is the Retinol-Binding Protein 4 a Possible Risk Factor for Cardiovascular Diseases in Obesity?"

_ijms, 2020, doi:10.3390/ijms21155229_

Round 1
Reviewer 1 Report
Comments to the Author
In this review article, the authors summarized the current knowledge about RBP4 and its association with essential aspects of cardiovascular disease-lipid profile, intima-media thickness, artherosclerotic process and diet, and RBP4 gene polymorphisms. Although it is well-written, some comments need to be addressed as follows:
- As in title name, RBP4 is a potential risk factor of CVD in obesity. Given that obesity is a low inflammation process with increased adipokines production and insulin resistance. Moreover, a review article describing that RBP4 is the causative factor and marker of vascular injury related to insulin resistance [Postepy Hig Med Dosw 2016;70:1267-75]. It would be better that the authors could add the related descriptions regarding association of RBP4 with other adipokines and insulin resistance.
- A recently published article reporting that the retinol, retinoic acid, and RBP4 are differentially associated with cardiovascular disease, type 2 diabetes, and obesity [Adv Nutr 2020;11: 644-666]. It would be better that the authors could have the related descriptions.
- Regarding this review, the authors could discuss the gender difference in the role of RBP4 in the pathogenesis of cardiovascular disease.
Reviewer 2 Report
The authors wrote a literature review on the urgent problem of studying the role of retinol-binding protein in the development of coronary heart disease in obesity.
Notes on the literature review.
1. A professional correction in English is required.
2. Correction of the list of references is necessary. Many references in the list of references do not indicate the year of publication and other output data (for example, in sources No. 16, 24, 26 and others).
3. It is necessary to increase the number of studies over the past 3 years (2018-2020).
After revision, the literary review requires a second review.
Round 2
Reviewer 2 Report
I see that the authors did a good job,the content of the article has been improved